# Comparison of Skin Prick Tests (SPT), Intradermal Tests (IDT) and *In Vitro* Tests in the Characterization of Insect Bite Hypersensitivity (IBH) in a Population of Lusitano Horses: Contribution for Future Implementation of SPT in IBH Diagnosis

**DOI:** 10.3390/ani13172733

**Published:** 2023-08-28

**Authors:** Vera Carvalho Pessoa, Manuel Branco-Ferreira, Sigridur Jónsdóttir, Eliane Marti, Paula Tilley

**Affiliations:** 1CIISA—Centre for Interdisciplinary Research in Animal Health, Faculty of Veterinary Medicine, University of Lisbon, 1300-477 Lisbon, Portugal; ptilley@fmv.ulisboa.pt; 2AL4Animals—Associate Laboratory for Animal and Veterinary Sciences, Faculty of Veterinary Medicine, University of Lisbon, 1300-477 Lisbon, Portugal; 3Immunoallergology University Clinic, Faculty of Medicine, University of Lisbon, 1300-477 Lisbon, Portugal; mbrancoferreira@gmail.com; 4Institute for Experimental Pathology, Biomedical Center, University of Iceland, Keldur, 102 Reykjavik, Iceland; sij9@hi.is; 5Department of Clinical Research and Veterinary Public Health, Vetsuisse Faculty, University of Bern, 3012 Bern, Switzerland; eliane.marti@unibe.ch

**Keywords:** horses, insect bite hypersensitivity (IBH), allergens, intradermal tests (IDT), skin prick tests (SPT), *in vitro* tests

## Abstract

**Simple Summary:**

Insect Bite Hypersensitivity (IBH) prevalence in Portugal and in Lusitano horses is not known, but environmental characteristics are favorable to the activity of *Culicoides* and common knowledge shows high occurrence in Lusitano stud farms. This study aimed to compare skin tests and *in vitro* allergy tests for the diagnosis of IBH in Lusitano horses. The results of our study showed that skin prick tests presented a higher discriminatory diagnostic potential in IBH diagnosis. This increases our knowledge about IBH in Lusitano horses and could represent a step forward in the future development of specific immunotherapy.

**Abstract:**

Thirty controls (C) and 30 IBH-affected (T) Lusitano horses were evaluated. T horses were included based on anamnesis and physical examination, supported by questionnaires. All horses were submitted to skin tests, Intrademal (IDT) and Skin Prick Tests (SPT), on the neck with 14 specific allergens, 13 recombinant proteins (r-proteins) from *Culicoides nubeculosus* (Cul n) and *Culicoides obsoletus* (Cul o) salivary glands and *Culicoides nubeculosus* Whole Body Extract (Cul n WBE). Addicionally, a cluster of six T and six C horses were also tested with Cul n 3 and Cul n 4 produced in insect cells and barley, as well as *E. coli* produced Cul o 3 and Cul o WBE. Allergen concentrations were 10 µg/mL for IDT and 100 µg/mL for SPT, and wheal diameters assessed at 20 min, 6 and 48 h. IDTs were considered positive when wheal diameter was ≥50% of the histamine wheal and SPT’s ≥ 0.9 cm. *In vitro* tests, allergen-specific serum IgE and sulfidoleukotriene (sLT) release assay were also carried out. Results showed that Cul n WBE, Cul n 7, 8, 9, Cul o1P and Cul o 2P were the best performing allergens for SPTs (*p* ≤ 0.0001) for the 1st allergen panel and Cul o WBE, Cul n 3 Bar and Cul n 4 Bac (*p* ≤ 0.05) for the 2nd, presenting a higher discriminatory diagnostic potential than IDTs, at a concentration of 100 µg/mL, with readings assessed at 20 min. Regarding *in vitro* tests overall, the sLT release assay performed best.

## 1. Introduction

The best characterized allergic disease in horses [1] is a type I hypersensitivity, IgE-mediated dermatitis caused by bites of hematophagous insects belonging to the genus *Culicoides* [2]. It is most frequently known as insect bite hypersensitivity (IBH) but can also be called *Culicoides* hypersensitivity, summer eczema or Queensland itch [3]. Insect Bite Hypersensitivity (IBH) is a recurrent, seasonal, pruritic dermatitis that affects many horses worldwide [4] and clinically manifests as a chronic relapsing seasonal dermatitis [4,5]. The prevalence of IBH varies between 3 to 60% depending on the environment and on the genetic background of the horse [3,6]. The prevalence of IBH in Portugal is not yet known. However, the environmental characteristics are favorable to the activity of the hematophagous insects most frequently involved in this dermatitis, which are present throughout the national territory [7,8]. As it is a multifactorial disease, involving genetic and environmental factors, as well as immune response mechanisms, a very variable prevalence is observed [4,9,10]. IBH diagnosis is still based on expert assessment comparing the presence of the representative lesions with the history of seasonal pruritus, mainly in the summer, and improvement of clinical signs in response to environmental management [4,11,12]. To date, the effective treatment for this disease remains elusive [3] and is based on insect control in the environment and use of topical and systemic antipruritic and anti-inflammatory agents (glucocorticoids, antihistamines, omega-6/omega-3 fatty acids). However, as it is an allergic disease, mostly IgE-mediated, there is the possibility of developing specific immunotherapy [4,13]. Various techniques are available to detect allergic sensitization, commonly designated as allergy tests. For several years, allergy diagnostic tests included Intradermal tests (IDT) and serological assays to determine allergen-specifc IgE [2], the latter using mostly ELISA techniques. IDTs using multiple allergens have been used as a diagnostic tool in IBH-affected horses for over 20 years [2,14]. A few studies have shown that IBH-affected horses more frequently have positive IDT results with *Culicoides* extract, and sometimes also with other insect extracts, than healthy control horses [3,4]. The involvement of *Culicoides* allergens in IBH has also been demonstrated with functional *in vitro* tests such as sulfidoleukotriene (sLT) release (CAST, Bühlmann laboratories AG) [15] or histamine release tests, using *C. nubeculosus, C.sonorensis* or *C. obsoletus* extracts [15,16,17], as sLT and histamine are more frequently released in IBH-affected horses than in healthy controls [3]. A previous study had shown a high correlation between *in vitro* sLT and histamine release in peripheral blood leukocytes from horses after a 40 min stimulation with *Culicoides* allergens [18], indicating that the sLT release can be used to detect immediate type I reactions *in vitro* to given allergens. However, as stated above, even though intradermal skin testing is still currently used to identify the allergens responsible for the disease, sensitization to a particular allergen does not necessarily mean that the individual is allergic or will develop clinical signs of allergy [2]. Other authors [19] previously published a study introducing Skin Prick Tests (SPT) as a valid diagnostic tool for the detection of positive reactions to environmental allergens in horses showing clinical signs of Equine Asthma Syndrome, as an alternative and/or complement to IDTs. Also, despite *in vitro* measurement of specific IgEs presenting in general a high specificity, they often present a lower sensitivity than SPTs in the diagnosis of IgE-mediated reactions [20,21]. Advances in the identification of the allergens involved in IBH provide the basis for both improved diagnostic methods with specific allergens and for new promising tools for targeted allergen immunotherapy (AIT) using the major allergens of *Culicoides* spp. [4,11,12,22,23,24].

In the present study, we evaluated the use of *in vivo* skin tests, SPT and IDT, and *in vitro* allergy tests, namely allergen-specific serum IgEs and sulfidokeukotriene (sLT) release assay in the diagnosis of IBH in Lusitano horses, using recombinant *Culicoides* allergens (r-*Culicoides* allergens) and *Culicoides* extracts. The present study contributes to increase our knowledge about IBH in Lusitano horses and could represent a step forward in the future development of specific immunotherapy.

### 1.1. Relevance of the Study

To the authors’ best knowledge, there are no studies about the impact of IBH in Lusitano horses, either in mainland Portugal, or outside. Nevertheless, Portugal presents all the environmental conditions for *Culicoides* spp. activity, and in the last few years, there has been an increase in IBH cases. Lusitano horse breeding plays an important role in the Portuguese economy and riders are making increasing use of the Lusitanos around the world [25]. IBH can result in the reduced commercial value of the affected horses, as well as extra costs to control the disease symptoms and altered behavior due to discomfort. Moreover, euthanasia may be considered when the symptoms are too severe and there is no way of controlling their impact on animal welfare. Furthermore, the results of this study may have an important contribution to the future implementation of locally relevant SPT allergen panels for IBH diagnosis and eventually for specific immunotherapy.

### 1.2. Aim of the Study

The present observational cross-sectional study aimed at characterizing and comparing the results of skin allergy tests, both IDTs and SPTs, and *in vitro* allergy tests (sLT release assay and serum-specific IgEs), in Lusitano horses from stud farms in Portugal with a clinical diagnosis of IBH, as compared to healthy controls living on the same farms.

## 2. Materials and Methods

### 2.1. Horses (Sample)

In total, 60 horses were tested, divided in two groups for the 1st allergen panel:Test Group (T)—30 IBH-affected horses presenting symptoms at the time of the tests;Control Group (C)—30 healthy horses.

For the 2nd allergen panel, a cluster of these horses, 12 animals (6 T and 6 C horses) in total, were simultaneously (1st and 2nd panels) tested in the same conditions, as described above.

#### 2.1.1. Characterization of the C and T Horses

(a)Ages

The mean age of the T and C groups was 7 and 9 years with a standard deviation of 5.32 and 6.06, respectively. The horses presented an age range that varied between 2 and 25 years and 2 and 27 years, in groups T and C, respectively. They were grouped into several age ranges, namely, 2–5 years, 6–9 years, 10–14 years, 15–20 years and over 20 years, with the highest rate of horses occurring, in the case of the T group, in the age range of 2–5 years (57%) and 6–9 years (37%) in group C. The lowest rate of horses tested, for both groups, was ≥20 years with 3 and 7% in groups T and C, respectively.

(b)Sex

Regarding sex, 73% of the animals tested were females; a total of 44 mares, 27 and 17 from C and T groups, respectively. Only a third of the tested animals were males (*n* = 16) and mostly were thoroughbreds; a total of 14, and only two geldings (1 each in C and T groups). In Portugal, most of the Lusitano females are breeders and live in an outdoor regimen all year round, being more exposed to the *Culicoides* spp., thus, presenting more signs of IBH than the males that are kept mostly indoors and less exposed to the *Culicoides* spp.

#### 2.1.2. Inclusion Criteria for the Horses in the T Group

Must be ≥1 year old;Living predominantly outdoors;Must present a seasonal pruritic dermatitis;The lesions must be no less than grade I (broken hair on the mane and/or base of the tail), from at least the previous equivalent season;No glucocorticoid or antihistamine therapy within the two weeks prior to the tests was allowed.

#### 2.1.3. Exclusion Criteria for the Horses

Horse breeds other than Lusitano;Gestational mares;Horses that presented other skin diseases;Horses that presented systemic signs of other diseases.

### 2.2. Skin Allergy Tests

Two different types of skin allergy tests were performed in all animals: SPTs and IDTs. These were mainly performed during springtime (from March to June, between 2013 and 2016), corresponding to the onset of the symptoms for the IBH-affected horses. The readings of the papules’ diameters were assessed at 20 min, 6 and 48 h.

#### 2.2.1. *Culicoides* Allergens

In total, in a first stage, 14 specific allergens were tested on all horses, which included 13 recombinant proteins, all expressed in *E. coli* [9,23] from *Culicoides nubeculosus* and *Culicoides obsoletus* spp. salivary glands (Cul n 1 to Cul n 11, Cul o 1P and Cul o 2P), as shown in Table 1, and *Culicoides nubeculosus* (Cul n) whole body extract (WBE).

In a second stage of the study, six other allergens were tested, but only on 12 horses, 6 IBH-affected and 6 C-group: they included the allergens Cul n 3 and Cul n 4, expressed both in insect cells and barley (Cul n 3Bac, Cul n 3Bar, Cul n 4Bac, Cul n 4Bar) [5] and the Cul o recombinant allergen Cul o 3 as well as a Cul o WBE. The *Culicoides* extracts [23] and all recombinant allergens had been produced and purified as described previously [5,9].

#### 2.2.2. Performing the Tests

Horses were sedated with intravenous detomidine hydrochloride 0.01 mg/kg (Domosedan^®^ 10 mg/mL, Orion Corporation; Espoo, Finland) 5 min before testing.

A 20 × 40 cm rectangle was then clipped on the left side of the neck, to allow 4 lines with 8 inoculation points each, for SPTs. On the right side of the neck, a rectangle of 20 × 30 cm was clipped, allowing 3 lines with 6 inoculation points each, for the IDTs, as seen in Figure 1A,B.

Five cm distance was allowed between inoculation points. For the IDTs, 0.1 mL of each allergen was injected at a concentration of 10 μg/mL, using a 25GX5/8 needle. For the SPTs, a single drop of each of two concentrations (10 and 100 μg/mL) was inoculated per allergen (Figure 2).

Papule diameters were measured 20 min after inoculation and at 6 and 48 h. Saline, PBS or the buffer in which the recombinant allergens were soluble (Table 1) were used as negative controls, and histamine chlorhydrate (10 mg/mL) as the positive control. To avoid technical errors performing IDTs, all syringes were filled with the predetermined volume before starting the tests. All administrations were performed by the same investigator.

#### 2.2.3. Readings

Skin test results, for both IDTs and SPTs, were evaluated by directly reading the papules’ diameter. The mean of two orthogonal diameters (mm) was then calculated. For the IDTs, results were considered positive when the papule size was at least half the size of the histamine wheal (positive control), as reported previously [28]. For the SPTs results, a statistical analysis was performed to find the cut-off value for the wheal diameter to be considered positive. Cut-off values were evaluated from 0.8 to 1.2 cm, as is explained later in the results, and were considered as positive when the mean of the papules’ orthogonal diameters was ≥0.9 cm, with readings assessed at 20 min after inoculation.

### 2.3. Sulfidoleukotriene (sLT) Release Assay

To perform the *in vitro* sLT release assay, blood samples were collected into a vacuette containing ACD-B as anticoagulant (Greiner Bio One^®^, Kremsmünster, Austria) and shipped to the laboratory at the Vetsuisse Faculty, University of Bern, Switzerland within 24 h. The cellular antigen stimulation test (CAST, Bühlmann laboratories, Allschwil, Switzerland) was performed as described previously [15]. To briefly describe the technique: leucocyte-rich plasma was collected, transferred into a propylene tube, and then centrifuged. The supernatant was removed, and the pelleted cells resuspended in stimulation buffer (Bühlmann laboratories) containing heparin. Cells were then incubated in buffer only to determine spontaneous sLT release, or with anti-IgE (0.75 µg/mL) as stimulation control, and with whole body extracts (WBE) from *Culicoides* nubeculosus (Cul n WBE) (2 µg/mL) and *Culicoides obsoletus* (2 µg/mL). After 40 min, plates were centrifuged, and the supernatants transferred into 96-well microtiter tissue culture plate to be kept at −20 °C until assayed. Released sLTs were measured using the CAST ELISA (Bühlmann laboratories) following the manufacturers’ instructions. For all further evaluations, values of the net stimulation were used, i.e., the spontaneous sLT release was subtracted from values obtained with the anti-IgE or *Culicoides* WBEs. Data from horses with sLT values < 250 pg/mL after stimulation with the anti-IgE were considered as non-responders [18] and their results were not included for the analysis of the data obtained with the *Culicoides* WBEs. Three out of the 25 IBH horses and 4 out of the 24 controls tested in CAST were non-responders.

### 2.4. IgE Serology by ELISA

For the serological tests, jugular blood was collected from all horses into 9 mL vacutainer serum tubes, about 10 min before the onset of the skin tests, and serum was then separated by centrifugation. In agreement with previous studies [17,18,23,27,29,30], our serum samples were collected during *Culicoides* exposure season, so the serological responses were evaluated after the IBH-affected horses started showing clinical signs. Frozen serum samples were then sent to the laboratory of the Clinical Immunology Group, Vetsuisse Faculty of the University of Bern. Allergen-specific IgE was measured by ELISA as described previously [9,23], for the allergens listed in Table 1. ELISA plates were coated with *Culicoides* recombinant allergens at a final concentration of 2 ug/mL in 0.2 M Carbonate–Bicarbonate buffer, pH 9.4 (Thermo Scientific, Waltham, MA, USA) for 2 h at 37 °C. After washing in 0.9% NaCl, 0.05% Tween 20, non-specific binding sites were blocked with blocking buffer (PBS- 5% dried milk powder and 0.05%Tween^®^ 20, pH 7.4) for 1 h at 37 °C. Plates were then washed twice with wash buffer and the sera, previously diluted 1:5 in blocking buffer, added in duplicates to the ELISA plate and incubated overnight at 4 °C. After washing, a monoclonal antibody specific for equine IgE (3H10, diluted in blocking buffer to a final concentration 1 mg/mL) was added to the plates and incubated for 2 h at room temperature (RT) on a shaker. After a further washing step, an alkaline-phosphatase-conjugated goat anti mouse IgG with minimal cross reactivity to horse serum proteins (Jackson Immuno Research, West Grove, PA, USA) was added and incubated for 1.5 h at RT on a shaker. After final washes, plates were developed with 1.5 mg/mL phosphatase substrate (Sigma, St. Louis, MO, USA) in 10% diethanolamine (Fluka; Sigma-Aldrich, St-Louis. MO, USA), pH 9.8 and absorption measured at 405 nm after 2 h. After subtraction of the blank, OD values were corrected for differences between the plates coated with the same allergen. The correction factor for each plate was calculated based on the average of the OD values of the positive controls included on all plates [5]. Results are shown as corrected OD values.

### 2.5. Statistical Analysis

Statistical analysis was carried out using the SPSS program (IBM SPSS Statistics 23) and different statistical tests were used to evaluate the skin allergy tests’ data (size of the papule), namely ANOVA Repeated Measures and Discriminant Analysis with a classification matrix ≥70% for SPTs and ≥60% for IDTs, testing three independent variables: Exposure– two groups (control—C and IBH-affected—T), Time—over three time points (20 min, 6 and 48 h) and Type of test—Skin Prick Test (SPT) and Intradermal Test (IDT). To determine cut-off values, non-parametric tests were used, namely Kruskal-Wallis’s test to determine statistically significant differences between groups (T and C) at each time, and Friedman’s test to compare the evolution over time, considering *p*-value (*p* ≤ 0.05). Cut-off values were selected at a given specificity and at the highest accuracy possible.

As to the sLT release assay and the IgE serology, medians were used as the results were not normally distributed. The Mann-Whitney U test was used to compare median values of the IBH and control groups. To evaluate the performance of the tests, non-parametric Receiver Operating Curve (ROC) analysis was performed and calculations of the area under the empirical curve (AUC) were performed using PROC LOGISTIC (NCSS version 11). ROC Curves’ analysis was used to select the best cut-off values in the sLT release assay. Cut-off values were selected at the highest accuracy possible. The level of significance was set at *p* ≤ 0.05 for all comparisons performed.

## 3. Results

All horses showed a positive response to histamine (positive control), in both skin tests performed, as shown in Figure 3.

Regarding negative controls, in case of PBS, there were some positive reactions in both T and C groups. In case of SPTs, we had a total of nine horses that showed positivity from T group and two from C group. The papules’ mean diameters were 1.021 cm (T) and 0.886 cm (C) with a standard deviation of 0.8110 and 0.8636 for T and C groups, respectively, with a significance of 0.31 (*p* > 0.05). Regarding IDTs, we had a total of 16 IBH-affected horses and 18 from C group that showed positivity to PBS, with the papules’ mean diameter of 1.6 cm (T) and 1.56 cm (C), for *p* > 0.05. Despite having positive values in T and C groups for both SPTs and IDTs, no statistically significant differences were detected; hence, no interference in the results of the significant allergens diluted in PBS was considered.

Hypersensitivity reactions to the *Culicoides* allergens were evaluated immediately by means of both methods to estimate possible significant differences between T and C groups and contribute to the identification of the specific allergens involved in IBH.

## 3.1. Skin Tests:

### 3.1.1. Skin Prick Test (SPT)

For the first panel of allergens, after a discriminant analysis with a classification matrix ≥70, significant differences between populations of C and T horses were found for Cul n WBE, Cul n 7, Cul n 8, Cul n 9, Cul o1P and Cul o2P (Table 2).

Differences between C (*n* = 30) and T (*n* = 30) groups for these allergens are shown in Figure 4 in the graphic representation of Estimated Marginal Means, for SPTs.

For the second allergen panel, the allergens that presented statistically significant differences and better discriminated between groups C and T for both IDTs and SPTs were Cul o WBE, Cul n 3Bar and Cul n 4Bac (*p* ≤ 0.05). However, the number of horses tested was quite small. Papules’ diameters mean and standard deviation (σ) for these allergens are shown in Table 3.

In general, and for both allergen panels, the papules’ diameters of the IBH-affected horses (T) were statistically larger when compared to the control group (C), as seen in Table 2 and Table 3.

On the other hand, statistically significant differences between C and T groups were only observed when a concentration of 100 µg/mL (*p* ≤ 0.05) was used, with readings assessed at 20 min.

Regarding the cut-off value for SPTs, evaluations were made from 0.8 to 1.2 cm. For a cut-off of 0.8 cm, there was a high variability and no statistically significant differences were found between groups. Considering a cut-off of 0.9 cm, it was possible to determine statistically significant differences between test and control groups, at 20 min.

Over time, there was a decrease in papules’ diameter, with the maximum mean value occurring at 20 min, and no papules were present at 48 h. This is shown in Figure 5, for Cul n WBE, comparing SPTs and IDTs, for the T group.

Estimated Marginal Papules’ Means Measures (cm), over time, for both Skin Prick Test (SPT) and Intradermal Test (IDT), for IBH-affected horses (test group), for Cul n WBE.

For the rest of the significant allergens, the diameter of the papules performed identically over time for both skin tests, IDTs and SPTs, reflected in identical and, therefore, overlapping graphs.

Regarding the 1st allergen panel, it was also possible to determine that all the IBH-affected horses showed positivity for one to ≥ four of the statistically significant allergens. Furthermore, 70% of these T group horses presented positivity to at least four allergens (Figure 6).

Regarding the control group, 10 horses showed positivity to just one of the statistically significant allergens, and only two horses presented positivity to ≥four of the statistically significant allergens, but the majority, 40%, presented no positivity to any of the allergens (Figure 6).

In addition, all IBH-affected horses, without exception, were positive for the Cul n WBE allergen, and only six horses from the C group were positive to Cul n WBE (Figure 7). The second highest frequency of positive results in the T group occurred for Cul n 9 (Figure 7).

Regarding the 2nd allergen panel, results are shown in Figure 8. At least one of the IBH-affected horses was positive to each statistically significant allergen, and Cul o WBE presented two IBH-affected positive reactions. No positive reactions were seen in the C group for any of the statistically significant allergens.

### 3.1.2. Intradermal Tests

In case of IDTs, a classification matrix of 60% was determined and statistically significant differences were found for Cul n WBE, Cul n 8, Cul n 9 and Cul o2P (Table 2).

In comparison with SPTs regarding the second allergen panel, the allergens that presented statistically significant differences and better discriminated between groups C and T were Cul o WBE, Cul n 3Bar and Cul n 4Bac (*p* ≤ 0.05). Papules’ diameters mean and standard deviation (σ) for these allergens are shown in Table 3.

For both allergen panels, the papules’ diameters of the IBH-affected horses (T) were statistically larger when compared to the control group (C), as seen in Table 2 and Table 3.

The authors also observed that for the tested allergens, the mean papules’ diameters in the IDTs were superior to those observed in SPTs (Table 2), but we must acknowledge that the volume that was inoculated in each site for the IDTs was by far higher than the volume used for the SPTs.

Over time, there was an increase in papules’ diameter, with the maximum mean value occurring at 6 h, and a decrease at 48 h (Figure 5).

Regarding IDTs, readings at 20 min, 6 and 48 h are shown in Figure 9. There was an increase at 6 h with a significant decrease at 48 h after inoculation. This observation may also be seen in Figure 5.

### 3.2. sLT Release Assay

The median sLT release was significantly higher in IBH horses compared to controls with Cul n WBE (142 vs. 84 pg/mL, *p* < 0.05) and more clearly, with Cul o WBE (445 vs. 12 pg/mL, *p* < 0.0001), as shown in Figure 10.

However, when using the cut-off value of 340 pg/mL that had been established in an earlier study [15], the sensitivity with Cul n WBE was very low (36%) with a moderate specificity of 85%. The use of Cul o WBE resulted in a better sensitivity (64%) and specificity (90%), but the performance of the test was still lower than previously published [15]. An ROC analysis was thus performed, as seen in Table 4.

It confirmed the better performance of the assay when using Cul o WBE instead of Cul n WBE, as shown by the higher area under curve (AUC) of 0.892 for Cul o WBE compared to Cul n WBE (AUC 0.699). Decreasing the cut-off for Cul o WBE from 340 pg/mL to 250 pg/mL allowed an increase of the sensitivity to 73% at the same specificity of 90%. The best accuracy of the test was obtained using a cut-off value of 70 pg/mL and resulted in a sensitivity of 86% and a specificity of 85%.

### 3.3. IgE Serology

The results from the IgE serology are summarized in Table 5.

Statistically significant differences were found between T and C groups (*p* ≤ 0.05) for the following allergens: Cul n 3, Cul n 4, Cul n 10, Cul o 2, Cul o 1P, Cul o 2P and Cul o 3 (Figure 11).

The area under curve for the *Culicoides* allergens ranged from 0.659 to 0.773 and shows that Cul n3 and Cul o1P are the best performing allergens (BPA).

## 4. Discussion

Our experimental observational cross-sectional study allowed us to analyze if IgE-mediated reactions to certain *Culicoides* allergens could distinguish between allergic and non-allergic horses. Based on the results obtained in the skin tests of the present study, Cul n WBE, Cul n 7, Cul n 8, Cul n 9, Cul o 1P and Cul o 2P may potentially be the most relevant of the tested allergens for the study of IBH in Lusitano horses. These seem to be promising candidates for an SPT diagnostic test panel, involving allergens which were chosen based on the geographical area. To determine which combination of specific allergens would result in the best panel, discriminant analysis by the stepwise method was used with IBH status as the outcome, as also used in another recent study [24]. This showed that the allergens described above, Cul n WBE, Cul n 7, 8, 9, Cul o 1P and Cul o 2P, were the minimum number of allergens that truly discriminated between the Test (T) and Control (C) groups, regarding SPTs, with a classification matrix of the discriminant analysis ≥70%, as stated above. If some of these allergens were not included in the panel, the classification matrix would be lower. In case of IDTs, Cul n WBE, Cul n 8, Cul n 9 and Cul o 2P were the allergens that discriminated between the Test (T) and Control (C) groups, with a lower classification matrix of the discriminant analysis (≥60%). Cul o WBE, Cul n 3Bar and Cul n 4Bac for both SPTs and IDTs, although tested on a smaller number of horses, were apparently also relevant, and the authors believe that they should be tested on a larger number of horses before any recommendations are made. Interestingly, Cul n 3 and Cul n 4 did not seem relevant in the first group of horses. This could be explained by the fact that Cul n 3 and Cul n 4 expressed in *E. coli* were used in the first group of horses, while the same allergens expressed in insect cells and barley [5] were tested in the second group. Proteins expressed in *E. coli* are often found in inclusion bodies and need a refolding step. As they lack posttranslational modifications, because eukaryotic proteins expressed in *E. coli* often form inclusion bodies and cannot always be refolded correctly, it may decrease the allergen’s functionality [31]. In the present study, 70% of the IBH horses seem to be allergic to ≥ four of these allergens, in case of SPTs. The allergen that all IBH- affected horses were positive to was Cul n WBE. Although there are also some positive results for Cul n WBE in the C group (six animals), the wheal diameters for Cul n WBE were larger in the IBH-affected horses. Still, from our results, we could assess that Cul n WBE presented a specificity of 80%. It is known from previous studies [17] that IDTs using *Culicoides* whole body extract (WBE) often result in a positive reaction, even in clinically healthy horses. Natural allergenic extracts are heterogeneous and may contain non-allergenic components, in addition to allergens [32]. These extracts are susceptible to contamination with allergens from other sources and may contain enzymes proteolytic with the ability to reduce the concentration of allergen in the extract [32]. The associated problems to the use of natural extracts include difficulties with the assessment of potency and inconsistencies inherent in the production of extracts with equivalent contents of an allergen [32]. Almost all allergen sources contain multiple major and minor allergens, and even with the use of modern techniques, it is difficult to standardize these mixtures of different proteins [33]. As the Cul n WBE used in our study, in both IDT and SPT, was a whole-body extract (WBE) allergen containing in principle all allergens, this may explain why most of our IBH-affected horses and some horses from the C group showed a positive reaction to it. Nevertheless, the performance of the SPT with *Culicoides nubeculosus* WBE (Cul n WBE) was rather good. The specificity with some of the recombinant allergens was better, reaching 87%, but the sensitivity was usually much lower, at best 60% with Cul n 9. On the other hand, the use of recombinant allergens (r-allergens) may have an advantage compared to natural allergen sources, as they provide specific allergen products for the diagnosis and treatment of allergic diseases [34]. Recombinant allergens are produced with a higher degree of purity, in larger quantities, either with a similar capacity to bind IgEs when compared to their natural counterparts or with modifications to reduce the reactivity to IgEs (hypoallergens) [35]. However, as mentioned above, depending on the expression system used, their functionality may be reduced [31]. In humans, studies performed using r-allergens in skin tests revealed that their use is not only safe, but also has a good diagnostic efficacy when compared to natural extracts for diagnostic purposes [34]. Some previous studies in horses [5,17,26] suggested that the use of recombinant *Culicoides* allergens (r- Cul) may provide a more specific diagnosis of IBH in sensitized horses, decreasing the number of false positive reactions. Hence, the use of r-Cul allergens may allow better diagnosis and better allergen immunotherapy [5]. Our results showed that at least five r-allergens, Cul n 7, 8, 9, Cul o 1P and Cul o 2P, presented relevance for the study of IBH in Lusitano horses, as they have a considerably elevated diagnostic discriminant potential (≥70%) and seem to be a good choice to include in an SPT diagnostic test panel. New *Culicoides obsoletus* allergens [24] were discovered and produced after this study had been carried out, which may further improve the discriminant potential of the SPT for IBH. To the authors’ best knowledge, this was the first study comparing two different skin tests, IDTs and SPTs, *Culicoides* allergen-specific IgE determination and an sLT release assay in horses with a history of IBH, and specifically in Lusitanos. IDTs were, until now, considered the most sensitive and confirmatory skin tests for horses [16,24,26,36], and although some authors still argue in favor of IDTs [37], according to others [2], allergic horses may sometimes present several positive reactions to multiple allergens in IDTs, even to those allergens that are unlikely to cause allergies. Multiple studies over the years have shown that even healthy horses can react to IDTs [12,36,37,38], which can frequently induce false positive reactions in clinically healthy individuals [39,40]. This may represent an additional challenge to identify the allergens that truly induce disease. Also, IDTs may require a more specialized technique and interpretation of results may be more time consuming [13]. IBH is mainly an IgE-mediated immediate type I reaction [41,42] and skin test readings should thus be assessed at 20 min and at 6 h, when the binding of the allergen to mast cells, already loaded with specific IgEs, occurs. However, readings may also be assessed at 48 h, considering that later reactions may also occur [43], if there is cell involvement, or other late reactions due to the release of non-histaminic factors such as sulphidoleukotrienes (sLT) [15]. In SPTs, the amount of inoculated allergen is much smaller, when compared to IDTs. Nevertheless, extracts utilized for intradermal skin testing are less concentrated (for example 1:10–1:1000; 0.00001 μg/mL up to 1 μg/mL) than those used for SPTs, [44,45]. It is usual to use this difference in concentrations between IDT and SPT in human medicine, where SPT is indicated if a type I (immediate type) allergy is suspected, based on the medical history and clinical clinical signs [44]. Furthermore, previous work published by our group also used this higher concentration in SPTs in horses with asthma [19]. The technique used to perform SPTs is minimally invasive [44] and the probability of SPTs inducing false positive reactions is lower when compared to IDTs [44]. Hence, IDTs are considered less specific than SPTs according to other authors [44,45]. This observation is in accordance with our results, considering the classification matrix (≥70% for SPTs and ≥60% for IDTs) and C/T ratio determined for each allergen and test (IDT and SPT). For Cul n 7, Cul o 1P C/T ratio was ≥1 only for the IDTs, and no statistically significant differences were found between C and T groups. Hence, according to our results, SPTs presented a higher discriminatory diagnostic potential than IDTs in this population of studied horses and for the allergens tested. Furthermore, SPT results are immediately available and when carried out by trained professionals, can be interpreted in 20 min with no need for later readings. SPTs are easily reproducible at a relatively small cost, allowing the clinician to show a cutaneous reaction to a hard-to-convince owner [19,46]. Also, many different allergens can be tested simultaneously because the reaction to a specific allergen is localized to the immediate area of the SPT and systemic anaphylactic reactions are rare [44]. In our case, during the experiment, no anaphylactic reactions occurred. Furthermore, in human allergology, SPTs are an important cornerstone for standard diagnosis of type I, IgE-mediated, immediate allergic diseases [44,46] and wheal diameters above the positive control or ≥0.3 cm are considered positive [46]. In our case, a cut-off value of 0.9 cm was determined, and as seen in a previous study [19], where the cut-off value estimated was 1 cm, positive control values (histamine) were also much higher than those that normally occur in the human species. In the present study, results showed that SPTs performed at a concentration of 100 µg/mL, with readings assessed at 20 min, with a cut-off value of 0.9 cm, and presented better results than IDTs (10 µg/mL) in terms of discriminatory diagnostic potential. As IBH is known as a seasonal skin disease associated with a type I, IgE-mediated, hypersensitivity reaction to *Culicoides* spp. [47], the authors believe that further studies should be carried out to include SPTs in its diagnostic procedures. This would represent a step forward in the establishment of IBH diagnostic tools and allow an accurate determination of the allergens to which IBH-affected horses are sensitized. Nevertheless, even though the results found in the present study are quite promising, the clinical significance of specific IgEs and sLTs release assays in the diagnosis of IBH or any other allergic disease, shown by SPTs, should be interpreted according to the patient’s history and physical examination [40] and seen within the clinical context. In human allergy diagnostics, *in vitro* measurement of allergen-specific IgEs [48,49] remains an important complementary tool to diagnose type I, IgE-mediated, allergic diseases [44]. Skin tests may not be feasible, particularly if there are widespread skin lesions, even being contraindicated in situations involving the risk of anaphylactic shock. They may also sometimes lead to false negative or positive results [50,51]. The measurement of specific IgEs, despite presenting, in general, a relatively high specificity in the diagnosis of type I allergic reactions, often has a lower sensitivity than that of skin prick tests [20]. According to other authors [23], the diagnostic value of commercially available serological IgE tests for equine IBH is questionable [29]. Nevertheless, a previous study using ELISA has shown that IBH-affected horses have significantly higher serum IgE levels against recombinant *Culicoides* allergens than healthy control horses [9,24,26]. Hence, IgE levels to recombinant *Culicoides* allergens were determined in the sera of the C and T Lusitano horses. The T horses had significantly higher serum allergen-specific IgEs than the C group for Cul n 3, Cul n 4, Cul n 10, Cul o 2, Cul o 1P, Cul o 2P and Cul o 3. However, there was some overlap between the groups, meaning that some of the C horses also had serum-specific IgE for *Culicoides* r-allergens. This might be explained by a high degree of exposure to *Culicoides* bites resulting in some degree of asymptomatic sensitization [52]. A high exposure to allergens can lead to IgE sensitization in asymptomatic individuals due to some feedback mechanisms (for example, IgGs) that maintain the homeostasis of the immune system. Asymptomatic sensitization is defined as the presence of a positive skin prick test (SPT) and/or positive serum allergen-specific IgE in the absence of clinical allergic symptoms [53]. In humans, at least 10 to 20% of the population exhibit evidence of IgE-mediated sensitization and have never had relevant symptoms of allergic disease. Currently, there is no convincing explanation why some people with positive allergen tests do not show symptoms [53]. Cul n 3 and Cul o 1P were the allergens that better detected differences in allergen-specific IgE values between the C and T groups. Cul o1P also showed significant differences between T and C horses in the SPTs. Similarly, Cul o2P showed significant differences between groups, in both SPTs and IgEs ELISA. For Cul n 3, Cul n 4 and Cul n 10, there was no significant difference in SPT between T and C groups, although the T horses had significantly higher serum IgE in ELISA than the C group (Table 5). These discrepancies might be due to an incorrect refolding of the *E. coli*-expressed allergens, which is more important for functional tests such as SPT or cellular allergy tests than for binding of free serum IgE. Interestingly, barley or insect cell-expressed Cul n 3 and Cul n 4 led to significantly higher SPT reactivity in the T compared to the C group (Table 3). This illustrates the relevance of posttranslational modification and folding of the allergens for functional tests. Unfortunately, because of the lack of availability of the recombinant allergens, not all allergens could be evaluated both in skin tests and IgE ELISA. It has been previously stated [54] that *in vitro* measurement of allergen s-IgEs is the laboratory equivalent of clinical skin testing and could be an alternative to IDTs in horses, and equine practitioners often prefer to use only serological IgE assays to identify significant allergens [12]. The authors find that to determine the clinical significance of specific IgEs in IBH diagnosis, it is necessary to undertake SPTs and serum allergen-specific IgE testing, as well as considering the patient’s history and physical examination [40]. This agrees with previous studies in humans [40], and with our group’s study on equine asthma syndrome [19]. Importantly, serological IgE assays only show sensitization to an allergen, which does not necessarily reflect clinical allergy. Functional tests such as SPT are usually considered to be closer to the clinical expression of allergy. The same is true for functional *in vitro* assays. They are based on the *in vitro* activation of blood basophils in the presence of the allergen [54]. Some are based on the liberation of inflammatory mediators, such as histamine and sulfidoleukotrienes (sLts) (CAST—Cellular Allergen Stimulation Test) [50,51,55]. In some cases, the sulfidoleukotrienes (sLts) release assay by basophils stimulated by allergens (CAST) demonstrated greater sensitivity and specificity compared to other tests, including the determination of specific IgEs [55]. For equine IBH, CAST can so far only be performed with *Culicoides* extracts, as recombinant *Culicoides* allergens do not or only very weakly induce sLT release. Studies have shown that the CAST with *Culicoides nubeculosus* extract is useful to confirm the clinical diagnosis of IBH in horses from Switzerland [15]. The CAST results confirm the relevance of *Culicoides* allergens and of *Culicoides obsoletus*, for IBH in Lusitano horses. From all *in vitro* assays (IgE serology and CAST), the highest AUC was obtained for the CAST with *Culicoides obsoletus* extract (Cul o WBE) (AUC = 0.897, 95% Cl = 0.735–0.962), when compared with Cul n WBE. Recombinant *Culicoides* allergens were not used in the CAST because the *E. coli*-expressed allergen did not or only weakly induced sLT release (EM personal communication). Another limitation of the CAST is that fresh blood samples are needed. In conclusion, based on the results of this study for the SPTs, Cul n WBE, Cul n 7, Cul n 8, Cul n 9, Cul o 1P and Cul o 2P were the best performing allergens. Regarding *in vitro* assays, sLt release assay Cul o WBE was the best performing allergen, and even with a lower sensitivity and specificity, the serum allergen-specific IgEs, Cul n 3, Cul n 4, Cul n 10, Cul o 1P, Cul o 2P, Cul o 3 and Cul o 2, were determined as the best performing allergens in Lusitano horses.

## 5. Conclusions

The results of this study support the use of SPTs as a major contributor to the diagnosis of IBH and may represent a step forward as an IBH diagnostic tool and eventually in the establishment of patient-tailored, component-resolved specific immunotherapy. Moreover, the identification of relevant *Culicoides* allergens involved in IBH in Lusitano horses using the allergen-specific IgE ELISA and the sLt release assay suggests that it will be worthwhile to test further *Culicoides obsoletus* recombinant allergens in SPT, probably increasing the performance of this test for the diagnosis of IBH. However, in agreement with other authors [21], even though conducting objective tests, both *in vivo* and/or *in vitro*, is important to confirm the clinical suspicion of allergy, a thorough anamnesis and clinical examination remains essential for the diagnosis of allergy. Finally, the evaluation of new pharmacological or immunotherapeutic options for IBH will benefit from valid skin testing with a specific allergen panel, which may also allow for the comparison of results among study populations.

## Figures and Tables

**Figure 1 animals-13-02733-f001:**
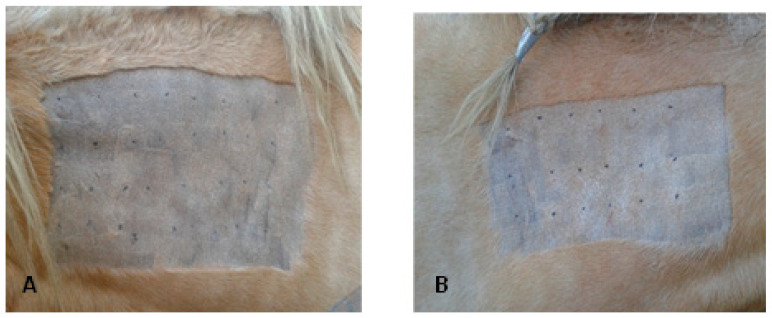
Skin tests performed on the horses’ neck; Skin Prick Tests (SPT) (**A**) were performed on the left side and Intradermal Test (IDT) on the right side (**B**).

**Figure 2 animals-13-02733-f002:**
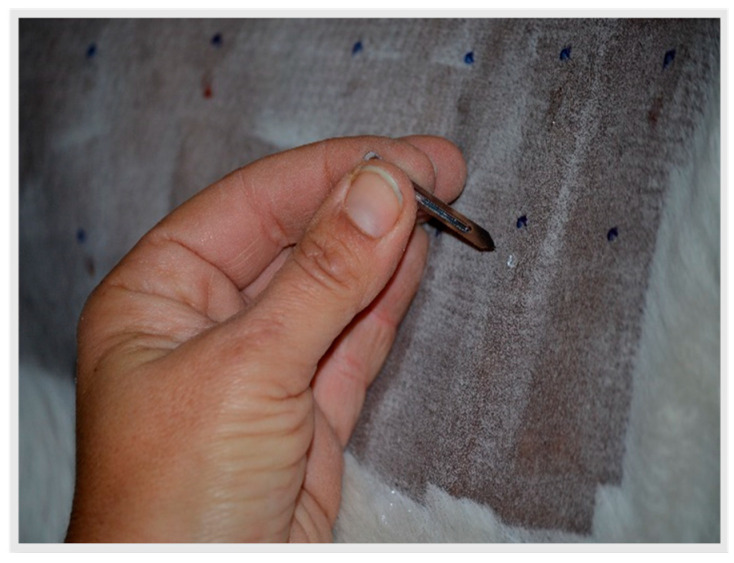
Skin Prick Tests (SPT) were performed by using a single drop of the allergen, placed on a marked site and a lancet was used for the inoculation.

**Figure 3 animals-13-02733-f003:**
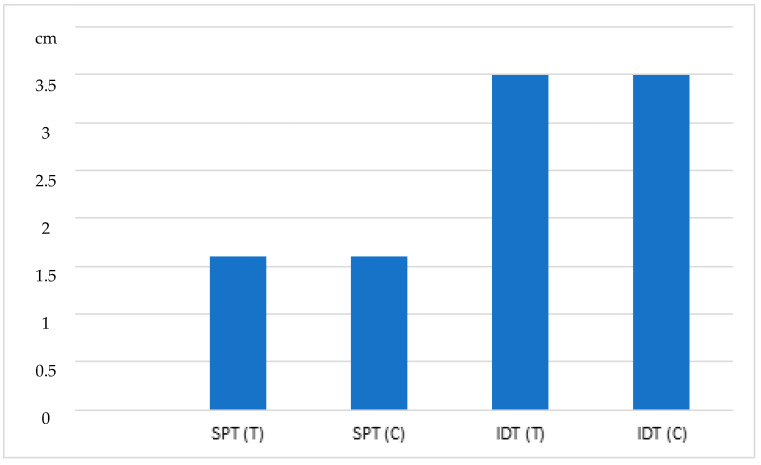
Absolute mean value of papules’ diameters (cm) for test (T, *n* = 30) and control (C, *n* = 30) groups evaluated in both skin tests (SPT and IDT) after inoculation of histamine (positive control). (σ: C = 1.2017; T = 1.1199, for IDTs). Abbreviations: T, test group; C, control group; IDT, Intradermal Test; SPT, Skin Prick Test; σ, Standard deviation.

**Figure 4 animals-13-02733-f004:**
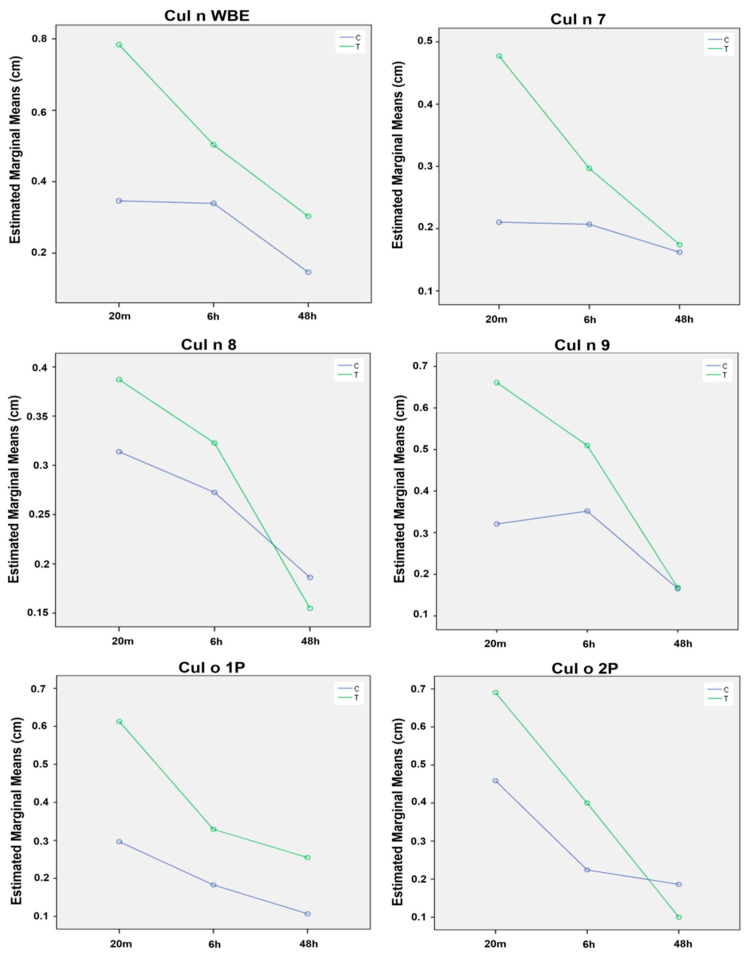
Estimated Marginal Papules’ Means Measures (cm) for the statistically significant allergens, over time, between IBH-affected (T, *n* = 30) and healthy (C, *n* = 30) horses tested, for the 1st allergen panel (SPTs, at a concentration of 100 µg/mL), with a classification matrix of 70%, (*p* < 0.05). Abbreviations: SPT, Skin Prick Test; T, Test group; C, control group.

**Figure 5 animals-13-02733-f005:**
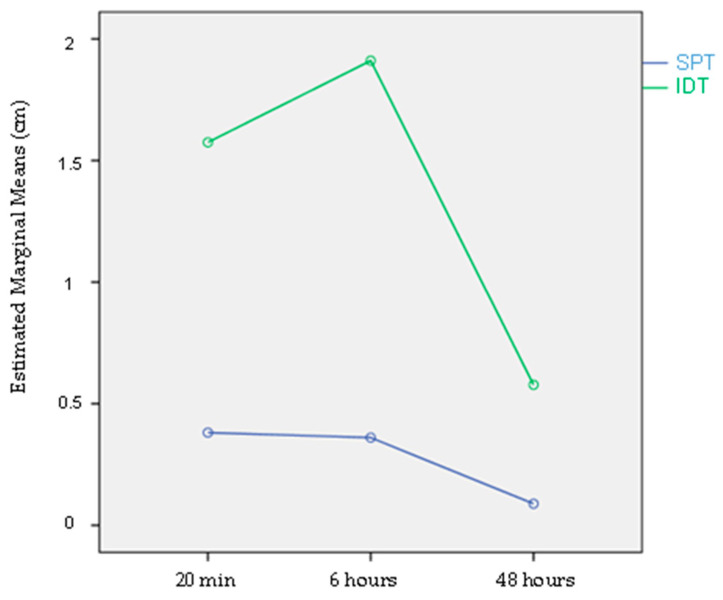
Estimated Marginal Papules’ Means Measures (cm), over time, for both Skin Prick Test (SPT) and Intradermal Test (IDT), for IBH-affected horses (test group), for Cul n WBE.

**Figure 6 animals-13-02733-f006:**
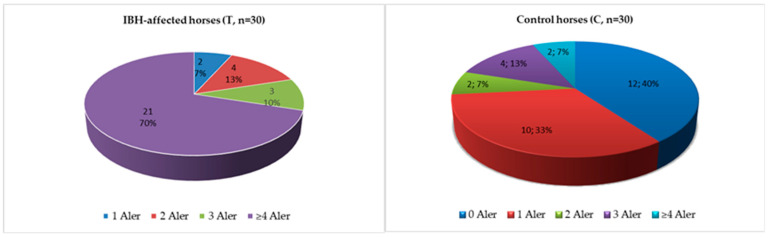
Total number of statistically significant allergens’ positive reactions for the 1st allergen panel, for both control (C) and test (T) groups, for SPTs, at a concentration of 100 µg/mL. Abbreviations: control (C); test (T); SPT, Skin Prick Test; Aler, allergens.

**Figure 7 animals-13-02733-f007:**
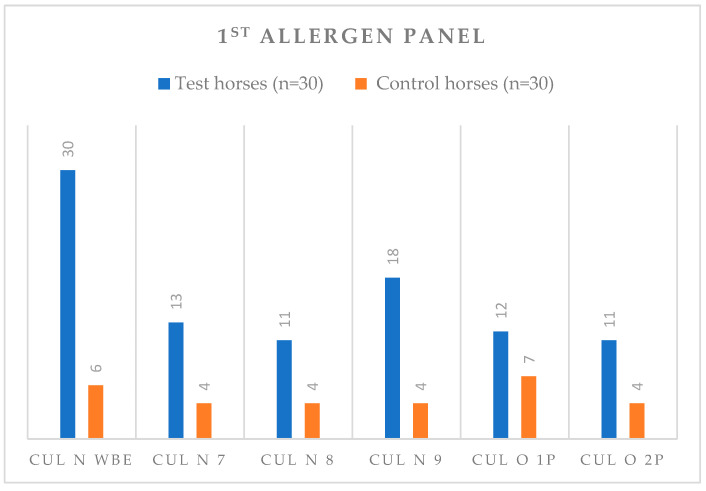
Number of horses with positive reactions for each statistically significant allergen tested (SPTs) at a concentration of 100 µg/mL, for both (C) and IBH-affected (T) groups, for the 1st allergen panel. Abbreviations: control (C); Test (T); SPT, Skin Prick Test.

**Figure 8 animals-13-02733-f008:**
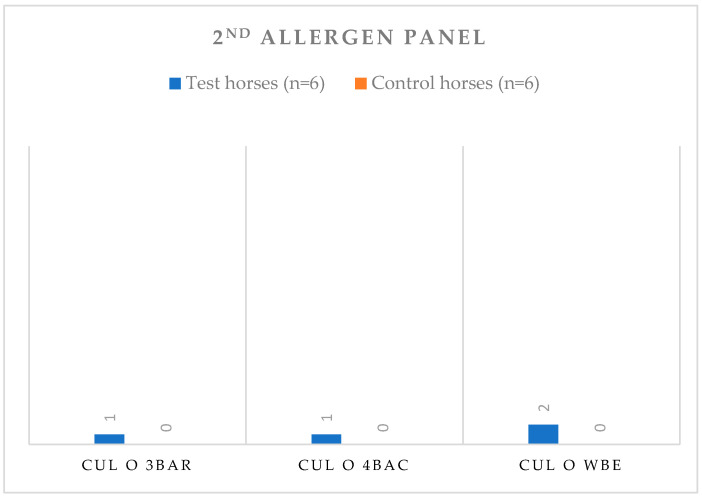
Number of horses with positive reactions for each statistically significant allergen tested (SPTs) at a concentration of 100 µg/mL, for both Control (C) and IBH-affected (T) groups, for the 2nd allergen panel. Abbreviations: control (C); test (T); SPT, Skin Prick Test.

**Figure 9 animals-13-02733-f009:**
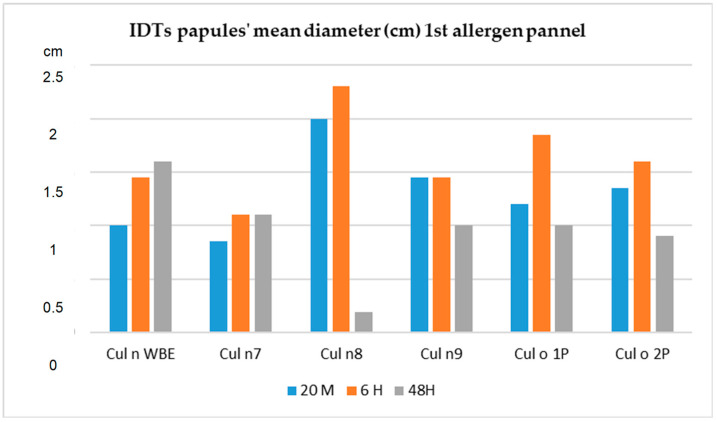
Intradermal test papules’ mean diameter (cm) over time Test group (T, *n* = 30), 1st allergen panel.

**Figure 10 animals-13-02733-f010:**
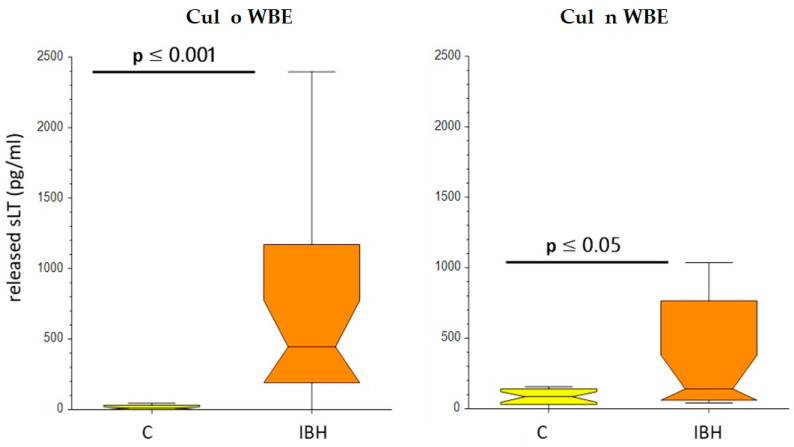
Sulfidoleukotriene (sLts) release assay with *Culicoides obsoletus* WBE (Cul o WBE) and *Culicoides nubeculosus* WBE (Cul n WBE). A total of 20 control (C) and 22 test (IBH) horses were tested. Results are presented as box plots, whereby the center horizontal line of the box plot marks the median of the sample. The edges of the box mark the first and third quartiles and the whiskers define the upper adjacent value, which is the largest observation that is less than or equal to the 75th percentile plus 1.5 times the interquartile range (IQR), and the lower adjacent value, which is the smallest observation that is greater than or equal to the 25th percentile minus 1.5 times IQR. *p*-values were calculated using the non-parametric Mann-Whitney U test (0.001 < *p* < 0.05). Abbreviations: Control (c, *n* = 20); test (IBH, *n* = 22).

**Figure 11 animals-13-02733-f011:**
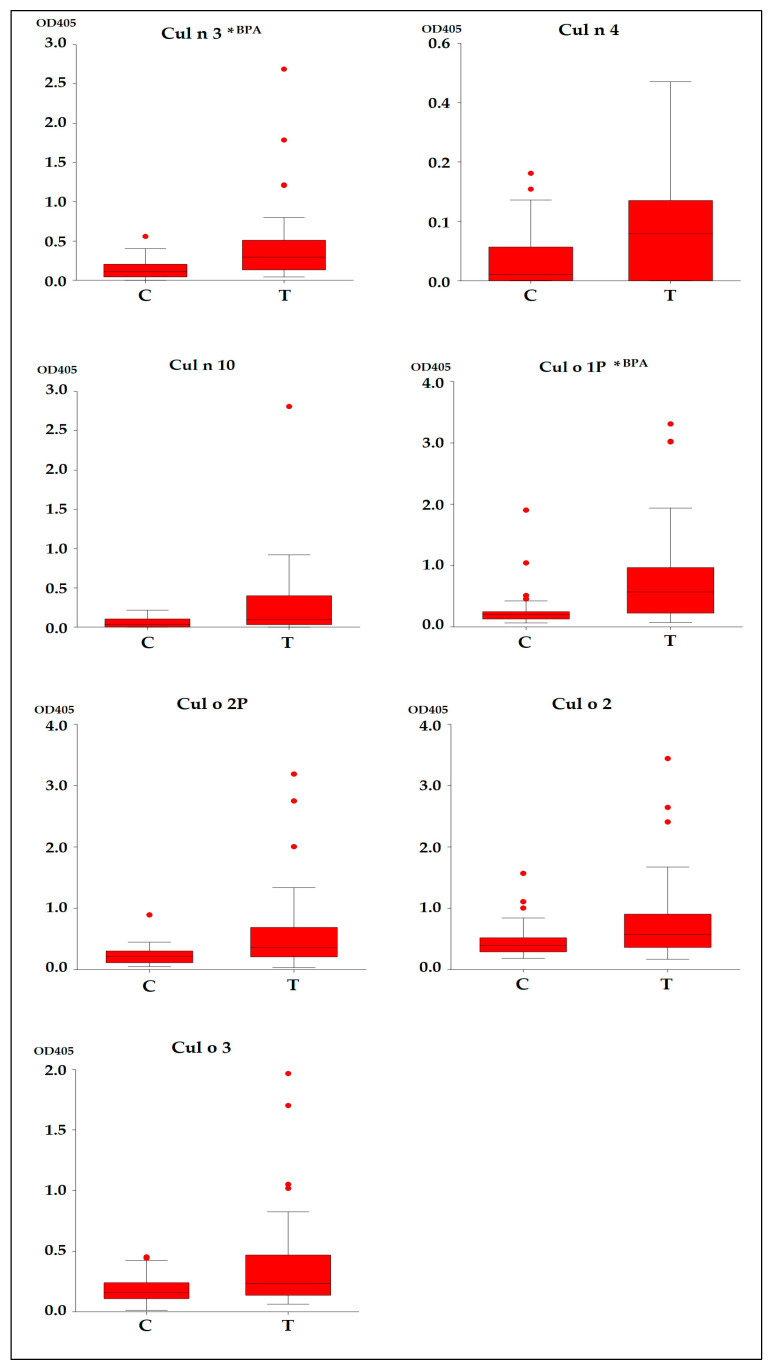
Boxplot representation for specific IgEs (OD405) that presented statistically significant differences, detected by ELISA in horses’ serum (*n* = 59). (*p* < 0.05). Abbreviations: C, Control Group, *n* = 29; T, Test Group, *n* = 30. *^BPA^ Best Performing Allergens.

**Table 1 animals-13-02733-t001:** Allergens used in Skin Tests and *in vitro* testing of IBH-affected horses (T) and control horses (C).

Allergen	Buffer	Skin Test (ST)/Serology (S)	References
Cul n 3	PBS	S/ST	Schaffartzik et al., 2011 [9]
Cul o 1P	20mM Tris 0.5M NaCl	S/ST	Peeters et al., 2013 [23]
Cul o 2P	20mM Tris 0.5M NaCl	S/ST	Peeters et al., 2013 [23]
Cul o 3	20mM Tris 0.5M NaCl	S	Van der Meide et al., 2013 [26]
Cul o WBE	0.9% NaCl	ST	Peeters et al., 2013 [23]
Cul n 1	H_2_O	ST	Schaffartzik et al., 2011 [9]
Cul n 2	PBS	ST	Schaffartzik et al., 2011 [9]
Cul n 3	H_2_O	S/ST	Schaffartzik et al., 2011 [9]
Cul n 4	H_2_O	S/ST	Schaffartzik et al., 2011 [9]
Cul n 5	H_2_O	ST	Schaffartzik et al., 2011 [9]
Cul n 6	H_2_O	ST	Schaffartzik et al., 2011 [9]
Cul n 7	H_2_O	ST	Schaffartzik et al., 2011 [9]
Cul n 8	H_2_O	ST	Schaffartzik et al., 2011 [9]
Cul n 9	PBS	ST	Schaffartzik et al., 2011 [9]
Cul n 10	H_2_O	S/ST	Schaffartzik et al., 2011 [9]
Cul n 11	H_2_O	ST	Schaffartzik et al., 2011 [9]
Cul n WBE	0.9% NaCl	ST	Ziegler et al. 2018, [27]
Cul n 4 Bar	PBS	ST	Jonsdottir et al., 2018 [5]
Culn 4 Bac	PBS	ST	Jonsdottir et al., 2018 [5]
Cul n 3 Bac	PBS	ST	Jonsdottir et al., 2018 [5]
Cul n 3 Bar	PBS	ST	Jonsdottir et al., 2018 [5]

**Table 2 animals-13-02733-t002:** Papules’ diameters mean and standard deviation (σ) for the statistically significant allergens (*p* < 0.05), for IDT and SPTs and both control (C, *n* = 30) and IBH-affected (T, *n* = 30) groups, for the 1st allergen panel and readings assessed at 20 min.

Allergen (1st Panel)	Test	Group	Mean (cm)	Standard Deviation (cm)	95% Confidence Limits	
					Lower	Upper
Cul n WBE	IDT *	C	1.47	0.45	1.308	1.631
		T	1.65	0.47	1.481	1.818
	SPT *	C	0.33	0.42	0.179	0.480
		T	0.78	0.54	0.586	0.973
Cul n 7	IDT ^NS^	C	1.12	0.55	0.923	1.310
		T	1.01	0.50	0.831	1.188
	SPT *	C	0.21	0.4	0.066	0.353
		T	0.48	0.5	0.301	0.658
Cul n 8	IDT *	C	0.87	0.57	0.666	1.074
		T	0.97	0.77	0.694	1.245
	SPT *	C	0.31	0.42	0.159	0.460
		T	0.39	0.45	0.228	0.551
Cul n 9	IDT *	C	1.21	0.58	1.002	1.417
		T	1.37	0.35	1.244	1.495
	SPT *	C	0.32	0.4	0.176	0.463
		T	0.66	0.57	0.456	0.863
Cul o 1P	IDT ^NS^	C	1.33	0.60	1.115	1.541
		T	1.30	0.62	0.803	1.521
	SPT *	C	0.30	0.40	0.156	0.443
		T	0.61	0.54	0.416	0.803
Cul o 2P	IDT *	C	1.21	0.73	0.948	1.471
		T	1.26	0.72	1.002	1.517
	SPT *	C	0.46	0.48	0.288	0.631
		T	0.69	0.45	0.529	0.851

Abbreviations: ^NS^ Non-significant allergens for IDTs; IDT, Intradermal Test; SPTs, Skin Prick Test. Significant differences are indicated by * (*p* < 0.05).

**Table 3 animals-13-02733-t003:** Papules’ diameters mean and standard deviation (σ) for the statistically significant allergens (*p* < 0.05), for IDT and SPTs, and both control (C, *n* = 6) and IBH-affected (T, *n* = 6) groups, for the 2nd allergen panel, and readings assessed at 20 min.

Allergen (2nd Panel)	Test	Group	Mean (cm)	Standard Deviation (cm)	95% Confidence Limits	
					Lower	Upper
Cul o WBE	IDT	C	1.46	0.33	1.195	1.724
		T	1.53	0.33	1.265	1.794
	SPT	C	0.13	0.33	−0.134	0.394
		T	0.65	0.51	0.217	1.082
Cul n 3 Bar	IDT	C	0.39	0.22	0.213	0.566
		T	1.56	0.29	1.327	1.792
	SPT	C	0	0	0	0
		T	0.48	0.54	0.0479	0.912
Cul n 4 Bac	IDT	C	0.53	0.29	0.340	0.762
		T	0.83	0.29	0.597	1.062
	SPT	C	0.23	0.36	−0.058	0.518
		T	0.27	0.47	−0.106	0.646

Abbreviations: IDT, Intradermal Test; SPTs, Skin Prick Test.

**Table 4 animals-13-02733-t004:** ROC analyses of the sulfidoleukotriene (sLTs) release assay with Cul o WBE and Cul n WBE (0.001 < *p* < 0.01).

			Z-Value to Test	Upper 1-Sided	95% Confidence Limits	
Allergen	AUC	Standard Error	AUC > 0.5	*p*-Value	Lower	Upper
Cul_n WBE	0.6986	0.0835	2.377	≤0.01	0.4965	0.8288
Cul_o WBE	0.8923	0.0533	7.363	≤0.0001	0.7262	0.9600

Abbreviations: AUC, area under curve; Control (C, *n* = 20); IBH-affected (T, *n* = 22) horses describing non-responders.

**Table 5 animals-13-02733-t005:** Results of allergen-specific IgE levels (OD Values) in horses’ sera determined by ELISA for both IBH-affected (T) and control (C) horses, and differences found for specific IgE levels between groups, T and C, using ROC Analysis (0.001 < *p* < 0.05).

Specific IgE Levels (ELISA)				ROC Analysis (*n* = 59)					
Allergen	C (N = 29)	T (N = 30)	*p* Value (U-Test)	Significance	AUC	Standard Error	Z-Value to Test	Upper 1-Sided *p*-Value	95% Confidence Limits
	Median (Min–Max)	Median (Min–Max)					AUC > 0.5		Lower/Upper
Cul n 3	0.106 (0.052–0.171)	0.293 (0.152–0.454)	*p* ≤ 0.001	Significant	0.7621	0.0616	4.251	≤0.0001	0.612/0.859
Cul n 4	0.016 (0–0.06)	0.119 (0.02–0.18)	*p* ≤ 0.05	Significant	0.6707	0.0702	2.432	≤0.0175	0.509/0.786
Cul n 10	0.035 (0–0.086)	0.09 (0.07–0.195)	*p* ≤ 0.05	Significant	0.7063	0.0674	3.060	≤0.01	0.548/0.815
Cul o 2	0.391 (0.323–0.444)	0.573 (0.439–0.776)	*p* ≤ 0.05	Significant	0.6747	0.0714	2.446	≤0.01	0.509/0.791
Cul o 1P	0.201 (0.167–0.217)	0.564 (0.26–0.82)	*p* ≤ 0.01	Significant	0.7730	0.0626	4.361	≤0.0001	0.495/0.778
Cul o 2P	0.216 (0.132–0.275)	0.36 (0.223–0.467)	*p* ≤ 0.05	Significant	0.7184	0.0672	3.249	≤0.001	0.559/0.826
Cul o 3	0.159	0.323	*p* ≤ 0.05	Significant	0.659	0.0719	2.221	≤0.01	0.495/0.778
	(0.114–0.201)	(0.159–0.389)							

## Data Availability

The data presented in this study are available upon request from the corresponding author. The data are not publicly available due to a privacy request from the horses’ legal owners.

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
