# Peer review of "Comparison of Skin Prick Tests (SPT), Intradermal Tests (IDT) and *In Vitro* Tests in the Characterization of Insect Bite Hypersensitivity (IBH) in a Population of Lusitano Horses: Contribution for Future Implementation of SPT in IBH Diagnosis"

_animals, 2023, doi:10.3390/ani13172733_

Round 1

Reviewer 1 Report

In general, please be consistent in italicizing Culicoides.

Please define WBE (whole body extract) and r- (recombinant) in the Abstract

Line 78 the reference should be 20, not 19, as 19 refers to sulfidoleukotriene and line 78 refers to skin prick tests.

Line 93 authors' , NOT authors

Line 146 I am unsure why the 'a' in all is italicized

Line 171 please check if reference 27 (Cordeiro Raposo A. O filho do vento) is the correct reference for the explanation of how positive IDT reactions were determined.

Line 192 and Line 194 I am unsure why WBEs is italicized

Line 247  'in case of PBS there were some positive...' NOT 'in case of PBS we had some positive...'

Table 2A Papules' NOT Papule's

Table 2B Papules' NOT Papule's

Figure 6, for IBH-affected horses, please delete the

Line 328 I think 'In comparison' is better than 'In accordance'

Line 330 Papules' NOT Papule's

Figure 8 Papules' NOT Papule's

Figure 9 Papules' NOT Papule's (in title, it is correct in legend.

Line 402-403 An English grammar rule is to not end sentences with a preposition. Say 'The allergen that all IBH- affected horses were positive to was Cul n WBE.

Line 411 In a scientific paper the word 'since' should only indicate the passage of time. Example 'Since the horse was 1 year old, the owners had noted pruritius'. In the case of line 411, use 'As the Cul n WBE used...'

Line 447 mast does not need to be capitalized

Line 451 'by far' is slang. Just use the word 'much'

Line 452-453 the statement 'the probability of SPTs inducing false positive reactions is lower when compared to IDTs.' should be referenced.

Line 490 insert the word 'than' between 'sensitivity' and 'that'

Line 500 'in' NOT 'to'

Line 507 separate 'no' and 'convincing'

Line 562 examination NOT exam

Line 598 Where is reference 1?

The English is very good. My few suggested changes are in the Comments and Suggestions to authors.

Author Response

Dear Reviewer:

Thanks in advance for your comments and remarks.

In annexe you may find a cover letter with the answer to your comments, to properly ellucidate your questions and remaks.

Best Regards

The authors

Reviewer 2 Report

I recommend critical reading of the manuscript by someone very proficient in English. Quite regularly words are chosen that mean something else then the authors most likely intended. This is somewhat confusing.

Author Response

(The authors gave the same response as above.)
